# Computational Fluid Dynamics Simulation on Thermal Performance of Al/Al_2_O_3_/SWCNT Nanocoolants for Turning Operations

**DOI:** 10.3390/nano12193508

**Published:** 2022-10-07

**Authors:** Vedant Joshi, Shardul Shrikhande, R. Harish, A. Giridharan, R. Mohan

**Affiliations:** School of Mechanical Engineering, Vellore Institute of Technology, Chennai 600127, Tamil Nadu, India

**Keywords:** turning operation, nanocoolants, cutting temperature, cutting fluid velocity, carbon nanotubes

## Abstract

The objective of this study is to numerically investigate the thermal performance of cutting fluids dispersed with nanoparticles for effective heat removal during turning operations. The simulations are performed using Ansys Fluent software, and the problem is modelled as a three-dimensional turbulent incompressible single-phase flow. The computational domain consists of a heated cutting tool and work piece, and nanocoolants are sprayed from a nozzle located above the machining zone. The nanocoolants are prepared by mixing mineral oil with nanoparticles of Al_2_O_3_ (Aluminium Oxide), Al (Aluminium) and SWCNT (Single Walled Carbon Nanotube). The heat transfer performances of different nanocoolants are compared by varying the nanoparticle volume fraction (φ) and coolant velocity (U_c_) in the range of 2% ≤ φ ≤ 8% and 1 m/s ≤ U_c_ ≤ 15 m/s, respectively. The results indicated a drastic drop in the cutting tool temperature with an increase in the volume fraction of dispersed nanoparticles and coolant velocity. The increase in volume fraction decreases the average cutting tool temperature by 25.65% and also enhances the average heat transfer rate by 25.43%. It is additionally observed that SWCNT nanocoolants exhibited a superior thermal performance and heat removal rate compared with Al and Al_2_O_3_ nanocoolants. The analysed numerical results are validated and are in good accordance with the benchmark results validated from literature.

## 1. Introduction

Many industries are using difficult to cut materials such as titanium and nickel based alloys in machining applications. The high heat generated during the machining process reduces the life of the cutting tool. Cutting fluids are being used in machining processes to evacuate the chips and to provide cooling and lubrication at the tool and workpiece interface. The tool temperature is the biggest factor affecting tool life and, hence, cutting fluid is sprayed at the tool and workpiece interface to maintain the optimal tool temperature. The utilization of nanocoolants for enhancing cooling performance during the machining process has wide applications in drilling, turning and milling operations. The addition of nanoparticles into the cutting fluid significantly increases the thermal conductivity of nanocoolants, improves the heat removal rate and further increases the life of cutting tool.

Machining can be improved significantly by using cutting fluids, as they offer higher cooling and lubrication. The minimum quantity lubrication system may be insufficient for turning the materials with outstanding hardness under high temperatures, such as M42 steel. In a case like this, nanoparticles are dispersed in the coolant to enhance the tribological properties and thermal conductivity of the base coolant. An experimental investigation by Anandan et al. [1] for turning AISI M42 steel using graphene nanofluids showed a notable decrease in temperature over oil and dried with minimum quantity lubrication conditions. A temperature reduction of 57% compared with oil and 82% with oil under a minimum quantity lubrication environment was noted. With the addition of nanofluid, the flank wear decreased by 95%, temperature by 82%, and surface roughness by 91% when compared to dry machining. The study conducted by Pal et al. [2] involves a minimum quantity lubrication technique with sunflower oil as the base fluid, which was further mixed with Al_2_O_3_ nanoparticles. The resultant colloidal dispersion was used for drilling of AISI 321 stainless steel. Under different cooling strategies, the drilling characteristics analysed were drill tip temperature, surface roughness, drilling forces and tool wear of AISI 321 stainless steel. They also found that dry drilling produced poor drilling characteristics results. In the case of 1.5 wt. % nanofluid minimum quantity lubrication conditions, the drilling forces, namely torque and thrust force, were approximately reduced to 82% and 40%, respectively, of that under flood drilling. The addition of Al_2_O_3_ nanoparticles plays an influential role by strengthening the cooling and lubrication properties of vegetable-based cutting fluids by improving the drilling characteristics with respect to the drill tip temperature. Moreover, nanofluid minimum quantity lubrication substantially lowers the intensity of adhesion on the tool compared to dry and flood conditions.

Das et al. [3] investigated the cutting performance and comparative evaluation towards improving machinability during the hard-turning operation of high-strength–low-alloy (HSLA) AISI 4340 steels using four different nanoparticles, namely CuO, ZnO, Al_2_O_3_, and Fe_2_O_3_, in deionized water by the minimum quantity lubrication technique. Results have shown that with the change in nanofluid from Al_2_O_3_ to CuO, the surface finish of the workpiece improves significantly. This improved surface finish is due to the lower viscosity of CuO base nanofluid as compared to ZnO, Al_2_O_3_, and Fe2O3 nanofluids. This further resulted in proper settlement of the nanofluid in the workpiece and tool interface, resulting in the cushioning effect, which generates lower vibrations and chattering during machining operation. Singh et al. [4] experimented with the surface grinding of Ti6Al4V-ELI alloy using a cubic boron nitride grinding wheel and analysed the tribological performance of three different nanoparticles, namely graphene graphite and MoS_2_ being dispersed in olive, soybean, and canola vegetable oils. The grinding performance of the synthetic fluid-based minimum quantity lubrication resulted in suitable performance. Jerold et al. [5] used cryogenic CO_2_ as a cutting fluid in the turning operations for machining AISI 1045 steel. The results showed that the use of cryogenic coolants reduced the cutting temperature by 5% to 22%. After reducing the cutting temperature, the tool life was observed to increase, since the adhesion and friction between the chip and the tool are reduced considerably. This improves chip breakability and surface finish. They identified that the cutting force is reduced between 17% and 38% using cryogenic machining. They also proposed cryogenic CO_2_ as an alternative to conventional coolants if environmental factors are monitored under cryogenic milling. Yi et al. [6] performed experiments to investigate the behaviour of graphene oxide nanofluids for turning Ti-6Al-4V. They investigated different parameters such as chip formation, cutting temperature and surface roughness. The results showed a reduction in cutting temperature from 35 °C to 30 °C when graphene oxide nanoparticles were used. Bouri et al. [7] conducted an experimental study to investigate the temperature profile near the cutting area. They concluded that aerosol water causes a higher rate of heat transfer and temperature reduction as compared to MQL. Furthermore, many studies on tool wear for turning applications have been conducted, and the influence of the cutting fluid has been analysed. Fang et al. [8] implemented a novel cooling channel for prolonging tool life and noticed that enhancing coolant pressure can improve the machining performance, thereby reducing the wear resistance. Gariani et al. [9] presented a comprehensive review on a novel cutting fluid impingement supply system which was developed to supply a decent amount of coolant to the machining zone, resulting in lesser cutting fluid consumption. The total reduction in cutting fluid was found to be 42%, and their novel design successfully rendered a 46.77% decrease in tool flank wear. Srikant et al. [10] proposed nanofluids as alternatives, mainly for minimum quantity lubrication, and have presented that nanofluids have an edge over other fluids due to improved thermal conductivity. The thermal conductivities make them suitable for applications in the machining industry. Furthermore, they [11] pointed out that 10% vegetable emulsifier oil can be used as an alternative to standard cutting fluids. Sharma et al. [12] carried out a review analysing various lubrication techniques including minimum quality lubrication, near-dry machining, high pressure pooling, cryogenic cooling and compressed cooling. Luchesi et al. [13] conducted experiments to investigate the heat transfer coefficients of cutting fluids. Ahmed et al. [14], in his research study, examined the Nusselt number and determined that an enclosure with square obstacles has a higher rate of heat transfer as compared to circular obstacles in buoyancy-driven flow. Other than nanofluids, several other techniques, such as Minimum Quality Lubrication (MQL), have been used for improving heat transfer. 

Choi et al. [15] proposed a high-performance heat transfer fluid developed by dispersing metallic nanoparticles into the base fluid. Coolants such as water, ethylene glycol or engine oil, with nanometric sized metallic particles when suspended, produce an enhanced engineering fluid with high thermal conductivity. Chummar and Harish [16,17] compared the thermal performance of water based nanofluid dispersed with different nanoparticles such aluminium, aluminium oxide, silver and single walled carbon nanotubes. The investigations were performed for natural and forced convection heat transfer by varying the Rayleigh and Reynolds numbers. It was found that the increase in particle concentration enhances the heat transfer rate. Murshed et al. [18] developed two static mechanism-based models to predict the improved thermal conductivity of fluids doped with spherical and cylindrical nanoparticles. They noticed that the thermal conductivity also increases with increase in temperature. Moreover, the thermal conductivity is also influenced by the size of the particle, shape and interfacial layer. Furthermore, Philip et al. [19] stated that the thermal properties of nanofluids are conflicting mainly due to the complexity of surface chemistry involved in nanofluids. They have also reported that nanoparticles are susceptible to agglomeration and settling. Their results on the thermal conductivities of oxide nanofluids using the transient-hot wire method proved that the addition of small quantity of nanoparticles increases the overall thermal conductivity of the colloidal suspension. Lee et al. [20] investigated on the thermal conductivity and heat transfer characteristics of nanofluid dispersed with of oxide nanoparticles. Bahmani et al. [21] investigated heat transfer, thermal efficiency and temperature variations in Al_2_O_3_/H2O nanofluid in counter and parallel flow heat exchangers. There is an enhancement of the average Nusselt number and maximum thermal efficiency in the counterflow regime that accounts to 32.7% and 30%, respectively. For a higher Reynolds number, adding more nanoparticles enhances the Nusselt number and coefficient of convective heat transfer. Wei Yu et al. [22] conducted an experimental investigation to decipher the thermal conductivity of graphene oxide sheet doped fluids. An effort was made to analyse the effective thermal conductivity with propyl glycol, distilled water and liquid paraffin as base fluids. It was found that the thermal conductivity of the effective fluid increases with an increase in the thermal conductivity of the graphene oxide nanoparticles, but decreases with an increase in the thermal conductivity of the base fluid. It has been proposed by the authors that this may arise due to the oxidation and defects of graphene. Anuar et al. [23] numerically investigated the flow and heat transfer characteristics of carbon nanotubes using H_2_O and kerosene as a base fluid. It was observed that with an increase in the volume fraction of carbon nanotubes, there is a decrease in skin friction and an increase in the rate of heat transfer. Hence, multi-walled carbon nanotubes are less efficient than single-wall carbon nanotubes in skin friction and heat transfer rate. Buongiorno [24], in his research study, has investigated seven slip mechanisms, due to which a relative velocity between the base fluid and nanoparticles exists. Accordingly, Brownian diffusion and thermophoresis are the most significant slip mechanisms in nanofluids. Lotfi et al. [25] conducted a study involving the analysis of forced convective heat transfer of nanofluids. It was reported that increasing the Reynolds number enhances forced convection, which, in turn, increases the Nusselt Number. Harish et al. [26] investigated the thermal performance of nanofluids undergoing thermal convection in a cubical enclosure and pointed out that the addition of nanoparticles increases the rate of energy exchange between the particle and fluid phase. They also observed that increasing the thermal conductivity of nanofluids intensifies the thermal convection within the fluid domain. Other than experimental studies, several comprehensive reviews have been undertaken by researchers to assess nanofluids for heat transfer implementations. 

Baghdadi et al. [27] conducted a study on a microchannel heat sink where nanofluids were used as coolants with the main purpose of increasing the heat dissipation rate. They considered SiO_2_, Al_2_O_3_ and CuO nanoparticles that are dispersed in water, and the results revealed that Al_2_O_3_ nanoparticles exhibited a higher heat dissipation rate as compared to others. The study also provided an important inference stating that the analysis of properties as a function of the Reynolds number is misleading, since different coolants have different thermophysical properties. With the addition of 5% concentration of carbon nanotubes in base fluid the skin factor and the heat transfer rate is improved by 26% [28]. They utilized computational fluid dynamics (CFD) software results to study the effectiveness of the nanofluid’s thermophysical properties. Bahiraei et al. [29] investigated the energy efficiency of a novel eco-friendly green graphene nanosheet nanofluid as well as its thermal and hydraulic features for mini heat sink applications. An increase in velocity or volume fraction improves temperature distribution uniformity and reduces the source temperature. In all the heat sinks, thermal resistance is reduced with an increase in the volume fraction of the nanoparticles and the velocity of nanofluids. Sadri et al. [30] performed a series of investigations in a heated stainless-steel pipe using clove-treated graphene nanoplatelets (CGNPs) nanoparticles suspended in distilled water under turbulent flow conditions. They validated the results and found that the numerical results were in excellent agreement with experimental findings. For evaluating the thermal performance of the nanofluids, the results proposed by the CFD model are highly accurate and reliable. Harish et al. [31] investigated the thermal performance of different nanofluids in a three-dimensional enclosure by taking into account the effects of thermophoresis and Brownian motion. Moreover, an increase in Reynolds number in inclined or horizontal geometry showed a significant increase in Nusselt number and heat transfer coefficient [32]. Daungthongsuk et al. [33] identified that the reasons for enhancement are due to disordered movement of nanoparticles, increased turbulence and fluctuations of the nanofluids which accelerates the energy exchange process. Oezkaya et al. [34] investigated the internal cooling conditions for drilling Inconel 718. They noticed that increasing the pressure generates an area of high flow rate near the cutting edge, which, thereby, results in greater heat transfer. Previous studies have also shown that classical molecular dynamics provide a better understanding of the behaviour in nanotube-material interfaces as compared to finite volume/difference/element methods. Gao et al. [35] conducted a molecular dynamics (MD) simulation to study the contact of a functionalized carbon nanotube (FCNT) with an immersed metal surface. Moreover, they studied the effect of thermal properties of FCNT nanofluid and, during rapid heating, noted the heat transfer. Molecular simulation highlights the intense influence on the heat transfer at the heating surface. Templeton et al. [36] developed a molecular dynamics model for including electric fields in the model using an atomistic-to-continuum framework, applying MD higher resolution of heat transfer interaction of CNTs with a metal surface.

However, most of the existing literature has not investigated the thermal performance of carbon nanotube coolants mixed with cutting fluids. Hence, the thermal performance of carbon nanotube coolants in comparison with other nanocoolants such as aluminium and aluminium oxide needs further understanding. The main objective of this investigation is to compare and contrast the thermal performance of different nanocoolants such as Al_2_O_3_, Al and Single Wall Carbon Nanotube (SWCNT) nanoparticles dispersed in the mineral oil coolant used in turning operations. The numerical investigation is performed using Computational Fluid Dynamics (CFD) to understand the temperature distribution on the surface of the cutting tool and workpiece for different nanocoolants during turning operation. The computational domain consists of a heated workpiece and tool, and nanocoolants are sprayed using a nozzle located above the machining zone. The parametric study is carried out by varying the nanoparticle volume fraction and nanocoolant velocity. The results are investigated by analysing the flow and heat transfer characteristics near the machining zone. Further investigations are performed by plotting contours of temperature, maximum cutting temperature, wall heat transfer coefficient and Nusselt number.

## 2. Mathematical Formulation and Governing Equations

### 2.1. Problem Statement and Physical Constraints

The computational domain of the turning process using the nanocoolant is shown in Figure 1. The problem is modelled as a three-dimensional turbulent incompressible flow. The nanocoolants are sprayed into the machining zone through a 5-mm diameter nozzle. The following assumptions have been considered:I.Viscous dissipation has been neglected for this particular study.II.The flow is considered an unsteady, three-dimensional incompressible turbulent flow.III.Nanofluid possesses the characteristics of a Newtonian fluid.IV.The solid nanoparticles of Al/${\mathrm{Al}}_{2}O_{3}$/SWCNT and the base fluid, which is mineral oil, are in thermal equilibrium. V.The effects of Joules heating and thermal radiations are neglected.VI.There is negligible potential for phase change.

### 2.2. Governing Equations

The cutting fluid is considered as mineral oil into which particles of Aluminium, Aluminium oxide and single walled carbon nanotubes are dispersed. The nanoparticles are considered to be monodispersed spherical particles. The governing equations of continuity, momentum, energy, turbulent kinetic energy and dissipation rate are given below.

Continuity equation:(1)$$\frac{\partial \left(\rho_{m}\right)}{\partial t}+\nabla \cdot \left(\rho_{m}\vec{V_{m}}\right)=0$$
where $\rho_{m}$ and $V_{m}$ constitutes the mass average density and the velocity of the nanofluid, respectively, and is formulated as follows:

Momentum equation:(2)$$\frac{\partial \left(\rho_{m\;}\vec{V_{m}}\right)}{\partial t}+\nabla \cdot \left(\;\rho_{m}\vec{V_{m}}\;\vec{V_{m}}\right)=-\nabla \;P_{m}+\nabla \cdot \;\left[\mu_{m}\;\left(\nabla \vec{V_{m}}+\vec{V_{m}^{T}}\;\right)\right]-\rho_{m}\beta_{m}\;g\left(T_{h}-T_{c}\right)\;\;\;\;\;$$
where the second term on the right-hand side represents the normal and shear stresses developed in the fluid particles, which is a viscous term. The last term couples the energy and momentum equation and formulates variation in density arising due to the difference between the hot and cold wall temperature.

Energy equation:(3)$$\frac{\partial \left(\rho_{m}C_{pm}T_{m}\right)}{\partial t}+\nabla \cdot \left(\rho_{m}C_{pm}T_{m}\vec{V_{m}}\right)=\nabla \cdot \left[k_{m}\left(\nabla T_{m}\right)\right]$$

Turbulence Model

The SST K-ω equation is used in this study, which comprises both the standard K-ε model and the Wilcox K-ω [37]. By substituting ε = Kω, the standard K-ε is transformed to K-ω [38]. Similar investigations [39,40,41,42] were performed to understand the capability of different turbulence models for predicting the heat transfer rate. These two equations are combined in order to apply the near wall treatment associated with the Wilcox model. In the inner layer, the sub-viscous layer effect is captured, along with the standard K-ω model, for capturing outer layer effects.

The SST K-ω model equations are given below:(4)$$\frac{\partial \left(\rho_{m}k\right)}{\partial t}+\frac{\partial }{\partial x_{i}}\left(\rho_{m}\vec{V_{m}}\;k\right)=\frac{\partial }{\partial x_{j}}\left[\Gamma_{k}\frac{\partial k}{\partial x_{j}}\right]+G_{k}-Y_{k}+S_{k}$$
(5)$$\frac{\partial \left(\rho_{m}\omega \right)}{\partial t}+\frac{\partial }{\partial x_{j}}\left(\rho_{m}\vec{V_{m}}\omega \right)=\frac{\partial }{\partial x_{j}}\left[\Gamma_{\omega }\frac{\partial \omega }{\partial x_{j}}\right]+G_{\omega }-Y_{\omega \;}+D_{\omega }+S_{\omega }$$

The generation of turbulent kinetic energy and the specific dissipation rate is represented as $G_{\omega }$ and $G_{k}$ respectively. $Y_{\omega }$ and $Y_{k}$ represent dissipation. $\Gamma_{\omega }$ and $\Gamma_{k}$ represent diffusivity. $S_{\omega }$ and $S_{k}$ represent source term. The blending function for the standard K-ε and the standard K-ω model is the extra cross diffusion term and is denoted by $D_{\omega }$.
(6)$$D_{\omega \;}=2\left(1-F_{1}\right)\rho \sigma_{\omega },\;2\frac{1}{\omega }\frac{\partial k}{\partial x_{j}}\frac{\partial \omega }{\partial x_{j}}$$

The Reynolds number and Nusselt number are evaluated using Equations (7) and (8), respectively.
(7)$$Re=\frac{\rho_{m}\;V_{m}\;D}{\mu_{m}}$$
(8)$$Nu=\frac{h\;l}{k_{m}}$$
where D and l represents the diameter of the nozzle and characteristic length of the tool tip.

## 3. Methodology

### 3.1. Selection of Suitable Nanoparticles—Thermophysical Properties of Nanofluid

In the present study, the thermophysical properties of different nanoparticles such as Al,$\;Al_{2}O_{3}$ and SWCNT and the base fluid mineral oil are tabulated in Table 1, Table 2, Table 3 and Table 4. The effective thermophysical properties of the nanofluid are obtained using the following equations. 

The effective density of the nanofluid is denoted by $\rho_{m}$ and is expressed as:
(9)$$\rho_{m}\;=\;\left(1\;-\;\phi \right)\rho_{bf}\;+\phi \rho_{np}$$
where ϕ represents the solid volume fraction of the nanoparticles. Furthermore, the thermal diffusivity of the nanofluid is given by:(10)$$\alpha_{m}=k_{m}/{\left(\rho c_{p}\right)}_{m}$$
where ${\left(\rho c_{p}\right)}_{m}$ denotes the heat capacitance of the nanofluid. It is determined by:(11)$${\left(\rho c_{p}\right)}_{m}=\left(1-\phi \right){\left(\rho c_{p}\right)}_{bf}+\phi {\left(\rho c_{p}\right)}_{np}$$

The thermal expansion coefficient ${\left(\rho \beta \right)}_{m}$ is expressed as:(12)$${\left(\rho \beta \right)}_{m}=\left(1-\phi \right){\left(\rho \beta \right)}_{bf}+\phi {\left(\rho \beta \right)}_{np}$$

In addition, $\mu_{m}$ symbolises the effective dynamic viscosity of the nanofluid, which is determined by:(13)$$\mu_{m}=\frac{\mu_{bf}}{{\left(1-\phi \right)}^{2.5}}$$

$k_{m},$ the thermal conductivity of the nanofluid, for spherical nanoparticles based on Maxwell model [35], is calculated using Equation (13):(14)$$k_{m}=k_{bf}\left[\frac{k_{np}+2k_{bf}-2\left(k_{bf}-k_{np}\right)\phi }{k_{np}+2k_{bf}+\left(k_{bf}-k_{np}\right)\phi }\right]$$
where $k_{bf}$ is the base fluid thermal conductivity, and $k_{np}$ is the thermal conductivity of dispersed nanoparticles. Moreover, the thermophysical properties of the base fluid and nanoparticles are shown in Table 1.

For the range of Al_2_O_3_ concentrations, the effective thermophysical properties are listed in Table 2. 

In Table 3, the numerical values of thermophysical properties for the SWCNT concentration range are listed and are used in the simulations.

In Table 4, the numerical values of thermophysical properties for the Al concentration range are listed and are used in these simulations.

### 3.2. Computational Domain 

The cutting tool and workpiece sample is modelled in Solidworks 2019. Both were modelled such that a small clearance existed between the cutting tool and workpiece. This was done to prevent bad mesh quality and infinitesimal mesh size at the interface. A fluid enclosure with a cushion of −X = 75 mm, +X = 50 mm, −Y = 15 mm, +Y = 50 mm, −Z = 225 mm, +Z = 50 mm is shown in Figure 1. In essence, the computational domain is modelled as a rectangular control volume with a length, breadth and height of 305 mm, 300 mm and 90 mm, respectively. A cylindrical nozzle of diameter 5 mm is modelled and is located above the machining zone. The nanocoolant sprayed from the nozzle impinges on the common surface of the cutting tool and workpiece. 

A single point cutting tool along with a cylindrical workpiece is modelled for simulation purposes. The diameter and length of the workpiece are taken as 40 mm and 200 mm, respectively. The cutting tool has a square cross-section with a side length of 25 mm. In order to reduce surface roughness and increase the cutting tool’s life, a nose radius of 0.67 mm is applied on the tip of the cutting tool. An end cutting edge and side cutting edge angle of 7.47° and 5.6°, respectively, are applied to avoid interference between the workpiece and tool. Additionally, a front clearance angle of 6.1° is considered on the cutting tool. Furthermore, a back rake angle of 6.52° and a side rake angle of 11.67° are considered on the cutting tool to facilitate a smooth flow of chips from the workpiece.

### 3.3. Numerical Discretisation

A tetrahedral mesh is deployed for the solid geometry and fluid domain, as shown in Figure 2 and Figure 3. Moreover, a body sizing is applied to the tip and main body of the cutting tool and workpiece. The tip and cutting tool/workpiece are given a body sizing of 0.15 mm and 1 mm, respectively. The mesh refinement is performed with very fine mesh near the tip of cutting tool and workpiece to evaluate heat transfer due to conduction. Since the fluid domain is not the main area of focus, a course tetrahedral mesh of size 10 mm is applied. The total number of nodes generated is 742,570, and the total mesh count is computed as 4.4 million cells.

### 3.4. Mesh Independence Study

To improve the disadvantages of the conventional method, the mesh count is varied from coarser to finer mesh until there is almost no variation in temperature concerning that of mesh count. The mesh count is iterated with 11 different mesh counts starting from 2.76 million mesh elements, which resulted in 0.58% of variation concerning the temperature. Then, the mesh is refined as fine, with a 4.39 million mesh count. This resulted in 0.23% of the variation in temperature, which is acceptable. Mesh skewness is 0.214, which lies in the range of excellent quality mesh. The skewness determines how close an ideal cell is, and skewness ranges from 0 to 1, where 0 is an equilateral cell and 1 indicates degenerated mesh. There is no rapid variation in the temperature value, as shown in Figure 4. The variations in the results were not significant and, hence, this mesh is considered for the calculation. This present mesh size prompted better results with less time when compared with finer mesh.

### 3.5. Numerical Validation with Previously Published Literature 

The present CFD results are validated with the experimental outcomes of Sharma et al. [12], and the comparison is shown in Figure 5. They studied the temperature distribution near the cutting tool area with Alumina (Al_2_O_3_) nanoparticles mixed with vegetable oil and water. Figure 6 displays the temperature contours on the workpiece and cutting tool after reaching a steady state. The distribution of temperature contours showed similar trends to that of the results observed by Sharma et al. [12]. Furthermore, Figure 5 shows that the temperature variation from the tool tip and the CFD results is close to the experimental benchmark results. The temperature drops from nearly 800 °C to 250 °C at a distance of 1 mm from the tip. Afterwards, the temperature attains a constant value near to a 100 °C. Although the constant temperature attained in this research study is slightly less than in the research study conducted by Sharma et al. [12], the curves follow a similar trend based on physics. 

### 3.6. Computational Fluid Dynamics Model

#### 3.6.1. Solver Settings 

The problem is modelled as an unsteady, three-dimensional turbulent incompressible flow. The temperature variations are created near the tool and workpiece tip due to conduction. Furthermore, parameters including convective heat transfer coefficient, Nusselt number and temperature are investigated. In addition, the k-*ω* SST (Shear Stress Transport) viscous model is used. The k-*ω* SST model comes under the most commonly used turbulence models to analyse the effect of turbulent flows. 

#### 3.6.2. Material Selection

The thermal properties of AISI 4130 Steel and M2 Molybdenum High Speed Tool Steel are shown in Table 5. The workpiece material is considered as AISI 4130 Steel, whereas the material of the cutting tool is treated as UNS T11302. A velocity-inlet boundary condition is applied to the nozzle cross-section, allowing the nanofluid to be sprayed on the cutting tool and workpiece. The bottom-most face of the domain is taken as a non-slip rigid boundary to emulate the bed of a lathe machine. The remaining boundaries are considered as pressure outlets, thereby allowing the cutting fluid to escape. The inlet velocity of the nanofluid is varied between 1 m/s and 15 m/s with step sizes of 5 m/s. This is primarily done to investigate the effect of Reynolds number on the heat transfer characteristics, such as the Nusselt number and convective heat transfer coefficient. As shown from Table 6, a temperature of 1003.15 K and 884.15 K is applied to the cutting tool tip and workpiece tip as an initial heat source for conduction.

#### 3.6.3. Solution Methods and Controls

The SIMPLE algorithm is used to avoid pressure–velocity coupling. A second-order upwind scheme is used for the turbulent kinetic energy and momentum equations. The energy equation and specific dissipation rates are also set to the second order upwind scheme. Moreover, the under-relaxation factors are modified for quicker convergence and accurate solutions. The residuals for mass, momentum, energy, turbulent kinetic energy and dissipation are set to a value of 0.0001.

## 4. Results and Discussions

### 4.1. Flow Evaluation of Temperature Contours with Respect to Time

The transient variation in temperature distribution of the tool and workpiece are shown in Figure 7. Accordingly, the temperature distribution on the rake and flank face of the cutting tool can be investigated. Furthermore, Figure 7 also illustrates that the temperature gradient increases with an increase in time, thereby indicating enhanced cooling. Furthermore, the temperature distribution stabilises after spraying the cutting tool–workpiece interface with nanofluid. A similar trend is observed in previously conducted research studies. Research conducted by Sharma et al. [12] involved the analysis of temperature distribution along the cutting tool using Alumina-MWCNT nanofluid. The contour bands observed in the form of circular arcs are analogous to those observed by Sharma et al. [12]. Furthermore, Bouri et al. [7] observed the heat transfer of MQL in turning operations. The temperature flow on the cutting tool follows a very similar pattern, with the maximum temperature of the tool tip being 932 K.

Figure 8a shows the temperature distribution when nanofluid flows over the tool and workpiece. The coolant velocity path from the nozzle to the tool–workpiece interference is shown in Figure 8b, depicting the perpendicular flow of nanofluid into the machining zone. Figure 9 visualizes the path traced out by nanofluid as it moves with the flow from the nozzle towards the tool–workpiece interference. This even spraying of coolant reduces the overall temperature due to friction and adhesion between tool and workpiece and increases the life of the cutting tool.

### 4.2. Temperature Distribution on Rake and Flank Face 

Figure 10 shows the temperature distribution on the rake and flank face of the cutting tool when nanofluid is sprayed. It can be noted from Figure 10e,f that the heat spread to the cutting tool body is less for SWCNT based nanofluids. Since the thermal conductivity of SWCNT is significantly greater than Al and Al_2_O_3_, the heat dissipation by the coolant is greater, thus resulting in less conduction for SWCNT based nanofluids. This trend can be observed on both flank and rake faces of the cutting tool. 

Figure 10 accurately demonstrates the decrease in temperature as the distance between the cutting tool and workpiece interface increases. From Figure 10a,b (i.e., Al_2_O_3_), the temperature at the bottom left corner is around 350 K. We can see a slight improvement in Figure 10c,d (i.e., Al), where the heat transfer is less as compared to the previous one. Figure 10e,f shows that the temperature reaches a constant value of 300.31 K at a lesser distance from the cutting tool interface when compared to Al and Al_2_O_3_. This indicates that SWCNT nanoparticles are more effective in cooling of the machining components. 

### 4.3. Effect of Nanoparticle Volume Fraction on Cutting Tool Heat Transfer Characteristics

A numerical investigation is performed for analysing the heat transfer and thermal performance of nanofluids for turning operations. Nanoparticles are doped in mineral oil for the purpose of enhancing thermal properties in order to improve heat transfer. The percentage of nanoparticles dispersed was varied at a constant velocity of 1 m/s. Hence, the Reynolds number of the flow is 55 and lies in the laminar region.

As noted earlier, increasing the volume fraction of nanoparticles doped in the fluid alters the overall properties of the colloidal suspension. This results in different heat transfer rates on the cutting tool and workpiece. It is observed that the thermal conductivity of the suspension increases significantly with the addition of nanoparticles. The volume fraction (Φ) of nanoparticles dispersed in cutting fluid is varied between 2% and 8%. Furthermore, it is observed from Table 2, Table 3 and Table 4 that the thermal conductivity increased with the increase in the volume fraction of nanoparticles. 

The maximum temperature attained by the cutting tool after the application of nanofluids is analysed based on different nanoparticles and distinct volume fractions. The volume fraction is varied between 2 and 8%. It is evident from Figure 11 that increasing the volume fraction results in a drop in maximum temperature. It is also observed that the increase in the volume fraction of nanoparticles in the suspension increases the thermal conductivity of the resultant fluid. This mainly occurs since the thermal conductivity of nanoparticles is very large as compared to that of the base fluid. The increase in the thermal conductivity of fluid results in better heat transfer and temperature distribution. Similarly, the specific heat of the resultant fluid decreases since less energy is required to cause a unit increase or decrease in temperature. As seen from Figure 11, it can be seen that the maximum temperature drops from 907 K to 899.5 K for all the nanofluid combinations. At all volume fractions in the range 2% ≤ Φ ≤ 8%, the temperatures of Al_2_O_3_ are lower when compared to SWCNT and Al nanofluids. Figure 11 simultaneously shows a linear decreasing trend from which we can conclude that a higher volume fraction of nanoparticles will result in a higher rate of heat transfer. Furthermore, the rate of decrease in temperature is higher for Al_2_O_3_ as compared to other nanofluids. In essence, we see that the temperature of the interface is at its maximum when nanoparticles are not dispersed in the base fluid. Similar findings on the capability of Al_2_O_3_ nanofluids are reported in literature [46].

The temperature gradient is defined as the change in temperature per unit length. Temperature gradient is a useful parameter which can be used to analyse the rate of change of temperature over a distance. As seen from Figure 12, the temperature gradient for a volume fraction of 2% is highest for SWCNT doped nanofluid. From Table 1, which shows the thermophysical properties of nanoparticles, it is evident that the thermal conductivity of SWCNT is 6600 W/m-K, which is more than 30 times greater than the thermal conductivities of Al_2_O_3_ and Al. Hence, the rate of heat transfer is greater for SWCNT doped nanofluid. Moreover, the temperature gradient decreases linearly with the increase in volume fraction. Furthermore, it can be seen from Figure 12 that the temperature gradient is nearly the same for Al and Al_2_O_3_ nanofluids at a volume fraction of 2%. The temperature gradient values of Al_2_O_3_ and Al nanofluids were found to be 2,729,600 K/m and 2,743,410 K/m, respectively. The temperature gradient on the cutting tool is observed to be slightly higher for the Al-mineral oil nanofluid. This trend is mainly observed since the thermal conductivity of Al is higher. Furthermore, the difference in the temperature gradient of Al–Mineral Oil and Al_2_O_3_–Mineral Oil nanofluids is greater at a higher volume fraction. The temperature gradient of Al–Mineral Oil nanofluid at 8% volume fraction is 2,708,530 K/m. On the other hand, the temperature gradient of Al_2_O_3_–Mineral Oil nanofluid at a volume fraction of 8% is 2,678,830 K/m, which is significantly lower.

The wall heat transfer coefficient between the nanofluid and cutting tool is analysed by varying the volume fraction of nanoparticles added to the cutting fluid. From Figure 13, the trend shows that increasing the volume fraction of nanoparticles results in a significant increase in the wall heat transfer coefficient. The convective heat transfer coefficient is defined as the ratio of heat flux to difference in temperature of the solid and fluid interface. Hence, the increase in volume fraction results in a larger heat transfer between the cutting tool and nanofluid. In addition, it is also observed that the wall heat transfer coefficient trend of the SWCNT nanofluid is higher as compared to Al and Al_2_O_3_, which is attributed to the fact that the specific heat of SWCNT is the least. Moreover, the wall heat transfer coefficient of the base nanofluid is less as compared to that when nanoparticles are dispersed. This validates the notion that nanoparticles are effective in enhancing heat transfer.

### 4.4. Effect of Reynolds Number on Cutting Tool Heat Transfer Characteristics

In the current research study, the effect of the velocity of the nanofluid on the heat source is studied under various parameters such as maximum temperature, temperature gradient and wall heat transfer coefficient. At higher temperatures, the tool life reduces drastically due to the massive heat generation caused by adhesion between the cutting tool and chips. A cutting fluid will act as a coolant and maintain optimum temperature, as the high-speed turning process will impact tool life and the workpiece’s surface by giving rise to microcracks, tensile residual stresses and thermal damage.

These problems in machining can be controlled mainly by reducing the cutting temperature. The conventional coolants fail to remove the heat effectively. To overcome this issue, a nanofluid with an increased volume fraction is used. In this study, the volume fraction is varied between 2% and 8%, and the best volume fraction is used to determine the effective velocity. It is interesting to note that with an increase in volume fraction, the average temperature is reduced. Furthermore, for a low Re, i.e., less than 200, the flow is under a laminar regime, and the nanoparticles are evenly distributed in mineral oil. At a higher Re, the flow is transformed into a turbulent flow. Accordingly, the Brownian motion and distribution of nanoparticles in mineral oil are highly chaotic at higher Re. However, the plot indicates that an increase in Re enhances the inertial force, and the temperature distribution is a decreasing function of Re, but substantial variation in the temperature distribution is not observed between different nanofluid pairs at higher Re.

The cutting fluid is varied in the range of 1 m/s to 15 m/s with three different Al, Al_2_O_3_ and SWCNT nanoparticles. Figure 14 shows the maximum temperature in terms of Reynolds number in three different nanofluids at the same 8% volume concentration of the nanoparticles, with cutting fluid velocities varying in the range of 1 m/s to 15 m/s.

In this numerical study, it is observed that with an increase in Reynolds number, there is a linear temperature drop in the curve. A similar trend was observed by Bahiraei et al. [29]. The nanofluid with Al_2_O_3_ nanoparticles showed better cutting temperature reduction than that of Al and SWCNT, as the Al_2_O_3_ curve lies below the curve of Al and SWCNT. Thus, we can conclude that there is a reduction in cutting temperature with an increase in nanocoolant velocity and Reynolds number. 

The trend of net temperature gradient is shown for different Reynolds numbers in Figure 15. It is evident from Figure 15 that the temperature gradient is at its maximum for SWCNT. At a constant volume fraction percentage, the thermal conductivity of SWCNT nanofluid is much greater than Al_2_O_3_ and Al. Furthermore, the temperature gradient decreases at a faster rate at a lower Reynolds number. This is attributed to the fact that the Brownian motion of the particles increases with the momentum of the fluid. Hence, we observe that the change in temperature gradient reduces with an increase in Reynolds number.

One of the most significant parameters for analysing the thermal performance of the nanofluids as a cutting fluid is the convective heat transfer coefficient. Figure 16 illustrates the convective heat transfer coefficient at various velocities. From Figure 16, it can be noted that with an increase in nanofluid velocity, there has been an enhancement in the convective heat transfer coefficient. Additionally, with an increase in the volume fraction of the nanoparticles, the convective heat transfer coefficient and Nusselt number increase. Figure 16 illustrates that SWCNT nanofluid has a greater convective heat transfer as compared to Al and Al_2_O_3_. This is valid for lower Reynolds numbers ranging from 55 to 432. On the other hand, for a Reynolds number higher than 432, it can be seen that Al_2_O_3_ represents the highest convective heat transfer coefficient when compared with Al and SWCNT. Hence, it can be concluded that SWCNT shows better heat transfer characteristics at a lower Reynolds number, whereas Al_2_O_3_ shows a higher convective heat transfer coefficient at Reynolds numbers greater than 432.

The Nusselt number is also an important criterion for analysing the thermal performance of a particular nanofluid. It is the ratio of convective and conductive heat transfer at a boundary in fluid. Figure 17 shows the variation in Nusselt number with respect to the Reynolds number for various nanofluids. It is evident from the graph that the Nusselt number for any given nanofluid increases as the Reynolds number increases. Furthermore, the Nusselt number increases at a higher rate when the Reynolds number is greater than 600. The Al_2_O_3_ dispersed nanofluids exhibit higher thermal performance than the Al and SWCNT dispersed nanofluids. 

## 5. Conclusions

In this research, a numerical investigation is performed using Computational Fluid Dynamics (CFD) to compare the temperature distribution on the surface of a cutting tool and workpiece during turning operation using different nanocoolants. The computational domain consists of a heated workpiece and tool, and nanocoolants are sprayed using a nozzle located above the machining zone. The nanoparticles of Al, Al_2_O_3_ and Single Walled Carbon Nanotube are dispersed in mineral oil for various volume fraction percentages. The heat transfer performances of different nanocoolants are compared by varying the nanoparticle volume fraction (Φ) and coolant velocity (U_c_). The parameters such as the maximum tool temperature, cutting tool temperature gradient, convective heat transfer coefficient and Nusselt number are investigated for different volume fraction and Reynolds number. The results indicated that nanofluid with higher thermal conductivity and lower specific heat capacity will exhibit better thermal performance. The conclusions from this investigation are summarized below: (a)The cutting tool and work piece temperature decreases linearly with an increase in the volume fraction of dispersed nanoparticles. This is attributed to the fact that a higher volume fraction increases the Brownian motion and thermophoresis of nanoparticles, resulting in greater thermal diffusion, which leads to drop in temperature.(b)The machining temperature is a decreasing function of Reynolds number. The increase in Reynolds number intensifies the inertial force of the nanocoolants and decreases the temperature in the machining zone.(c)The convective heat transfer coefficient increases linearly with an increase in the volume fraction of nanoparticles. The highest heat transfer co-efficient is observed for mineral oil dispersed with SWCNT nanoparticles.(d)The increase in particle volume fraction from 2% to 8% enhances the convective heat transfer co-efficient of mineral oil dispersed with SWCNT nanoparticles by 18.18%. Moreover, SWCNT nanocoolants exhibited the highest temperature drop per unit length with an increase in particle volume fraction compared to the other nanocoolant pairs.(e)The convective heat transfer coefficient and Nusselt number of nanocoolants are an increasing function of Reynolds Number. This is as a result of increased inertial forces, which accelerate forced convection heat rates in the machining zone.

It is evident from the present study that the forced convective heat transfer rate and the temperature drop in the machining zone are more significant for SWCNT nanocoolants than Al and Al_2_O_3_ nanocoolants. Hence, SWCNT nanocoolants exhibit superior thermal performance and are considered as the most suitable nanocoolants for turning operations. 

## Figures and Tables

**Figure 1 nanomaterials-12-03508-f001:**
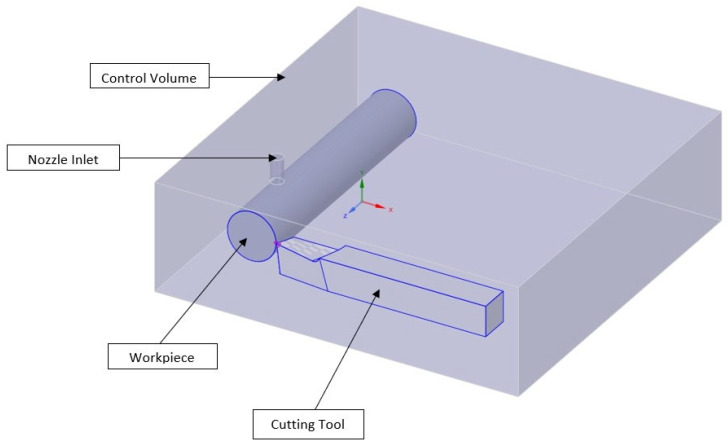
Computational domain comprising of cutting tool, workpiece and nozzle.

**Figure 2 nanomaterials-12-03508-f002:**
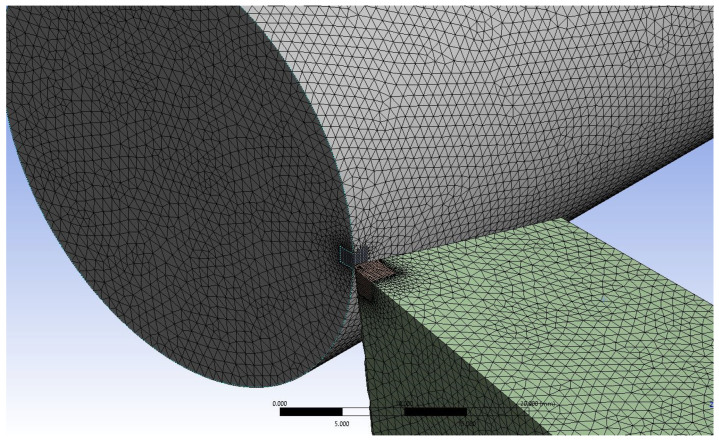
Meshed cutting tool and workpiece geometry with a mesh count of 2.45 million cells and an element size of 0.45 mm.

**Figure 3 nanomaterials-12-03508-f003:**
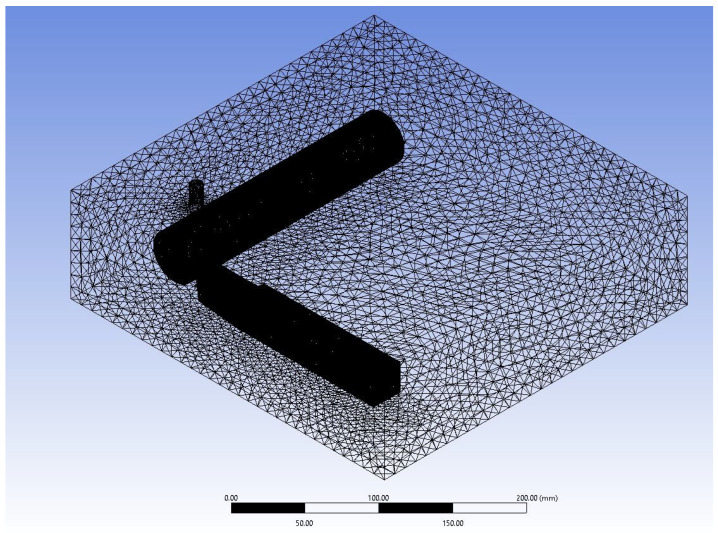
Meshed computational domain with a mesh count of 4.39 million cells and an element size of 5 mm.

**Figure 4 nanomaterials-12-03508-f004:**
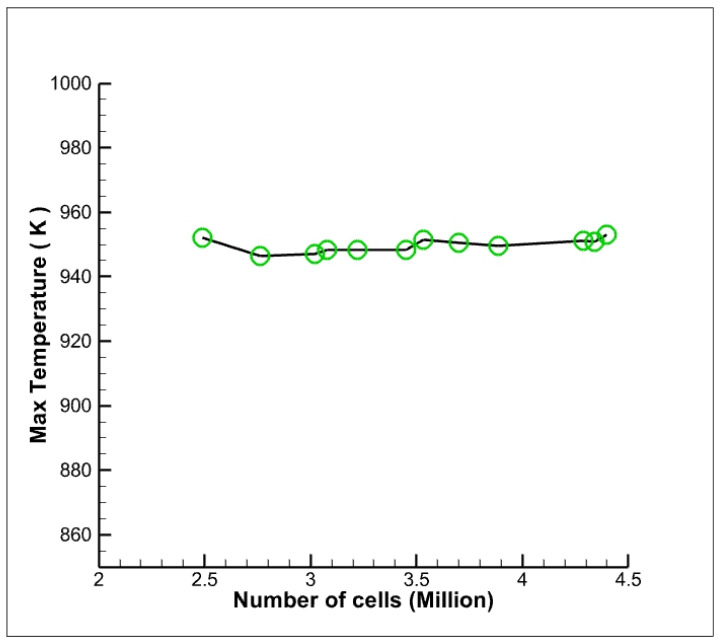
Grid independency test.

**Figure 5 nanomaterials-12-03508-f005:**
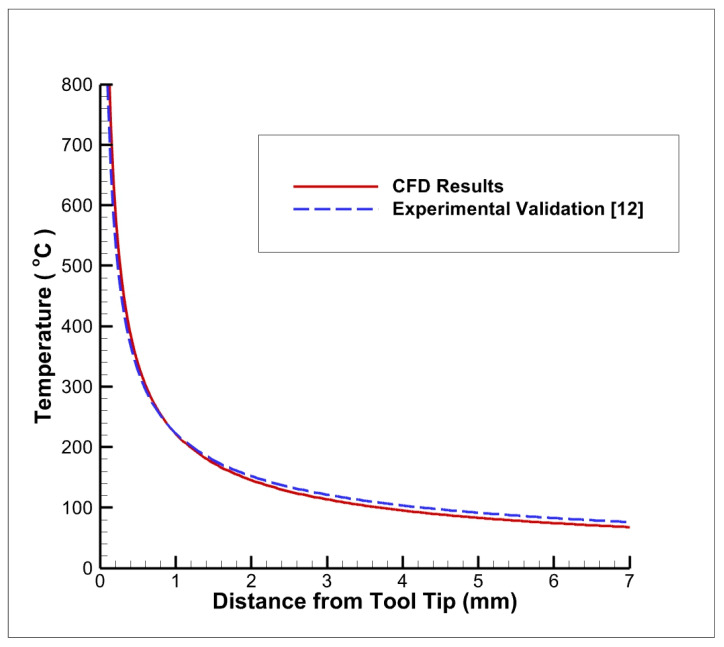
Numerical validation with benchmark experimental results [12].

**Figure 6 nanomaterials-12-03508-f006:**
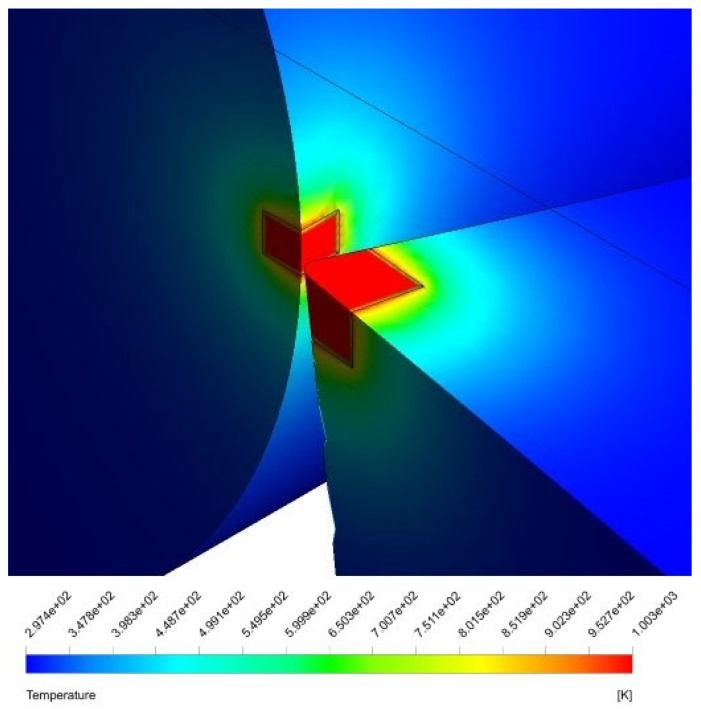
Variation in temperature with respect to distance from the cutting tool–workpiece interface.

**Figure 7 nanomaterials-12-03508-f007:**
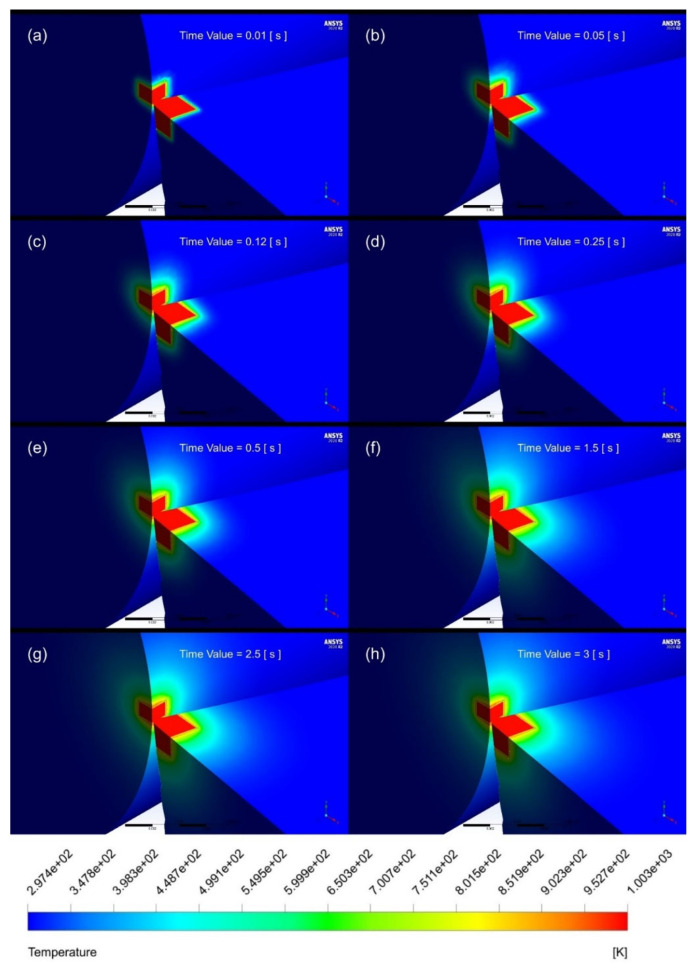
Temperature distribution at the cutting tool–workpiece interface using mineral oil at time (**a**) 0.01 sec, (**b**) 0.05 s, (**c**) 0.12 s, (**d**) 0.25 s, (**e**) 0.5 s, (**f**) 1.5 s, (**g**) 2.5 s and (**h**) 3 s.

**Figure 8 nanomaterials-12-03508-f008:**
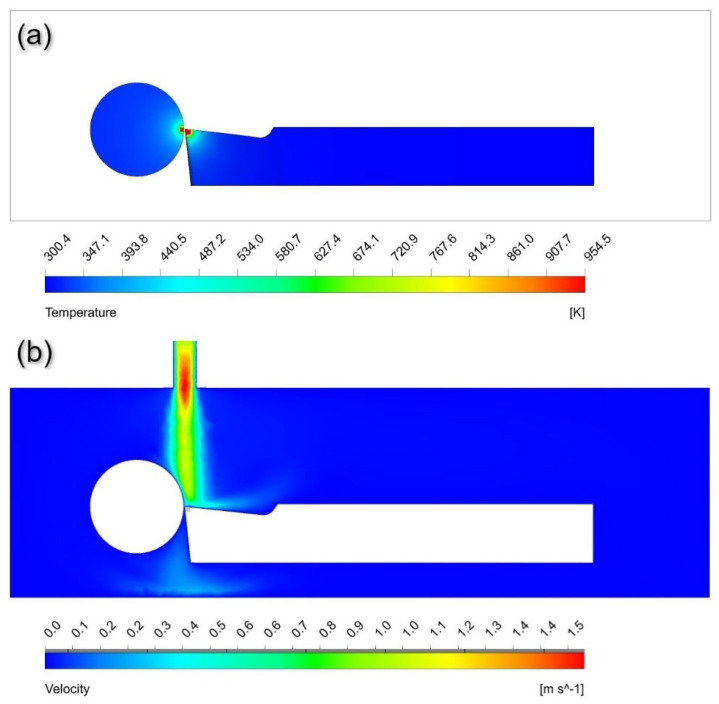
(**a**) Temperature distribution on the cutting tool and workpiece at steady state. (**b**) Velocity contours in the computational domain representing the flow of nanofluid on the machining interface.

**Figure 9 nanomaterials-12-03508-f009:**
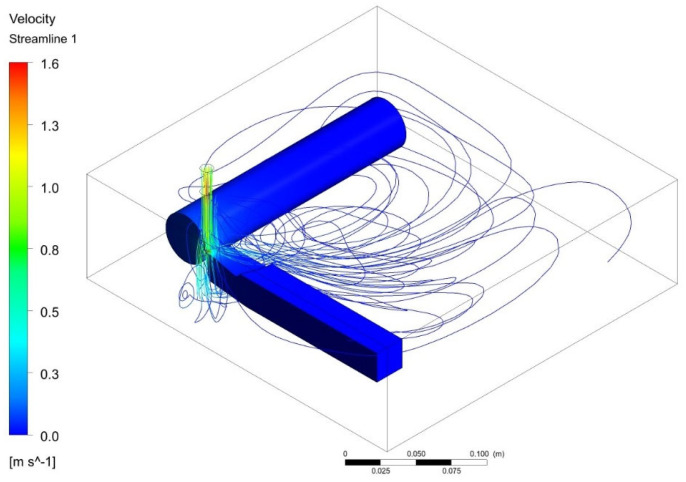
Streamlines depicting the flow of coolant from a nozzle to the machining zone.

**Figure 10 nanomaterials-12-03508-f010:**
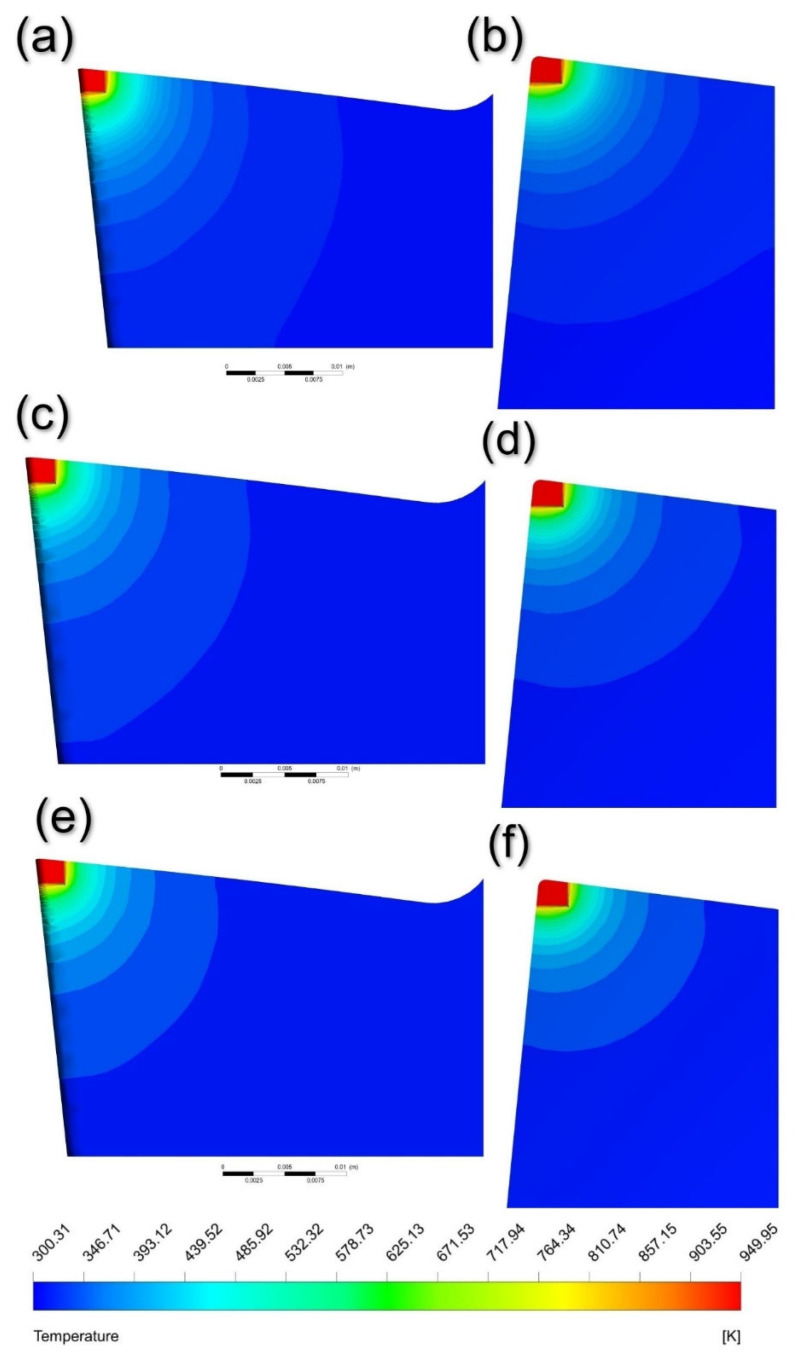
Temperature distribution of the cutting tool on (**a**) Al_2_O_3_—Flank Face, (**b**) Al_2_O_3_—Rake Face, (**c**) Al—Flank Face, (**d**) Al—Rake Face, (**e**) SWCNT—Flank Face and (**f**) SWCNT—Rake Face.

**Figure 11 nanomaterials-12-03508-f011:**
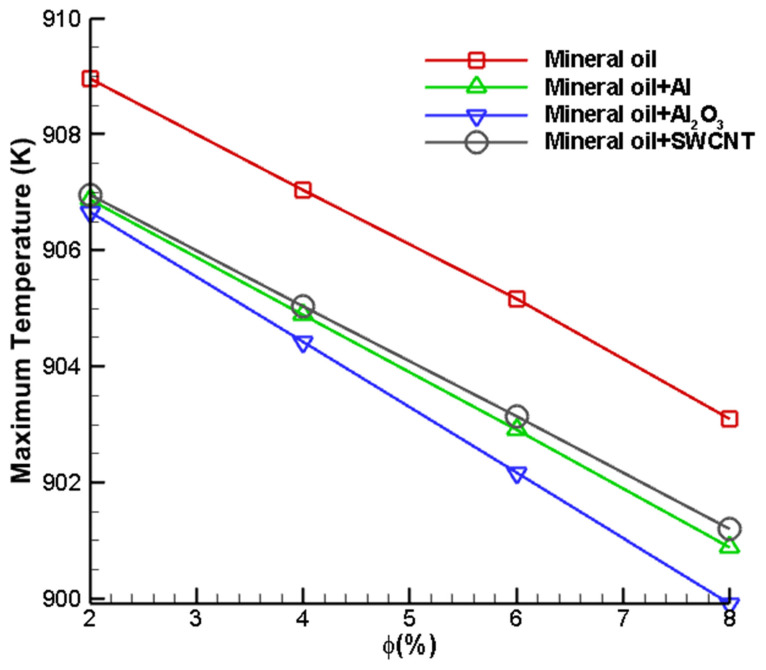
Variation in maximum cutting tool temperature with different volume fractions of dispersed nanoparticles.

**Figure 12 nanomaterials-12-03508-f012:**
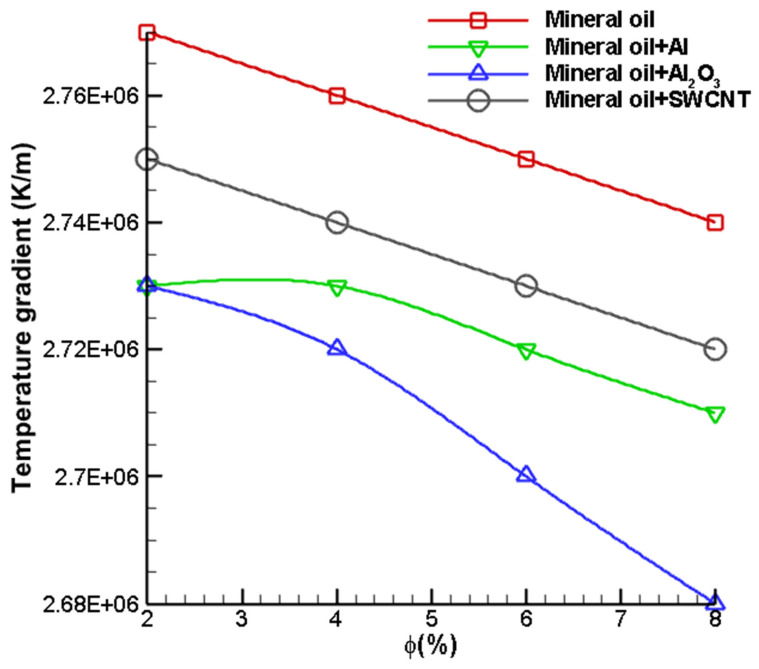
Variation in the temperature gradient on the cutting tool with different volume fractions of dispersed nanoparticles.

**Figure 13 nanomaterials-12-03508-f013:**
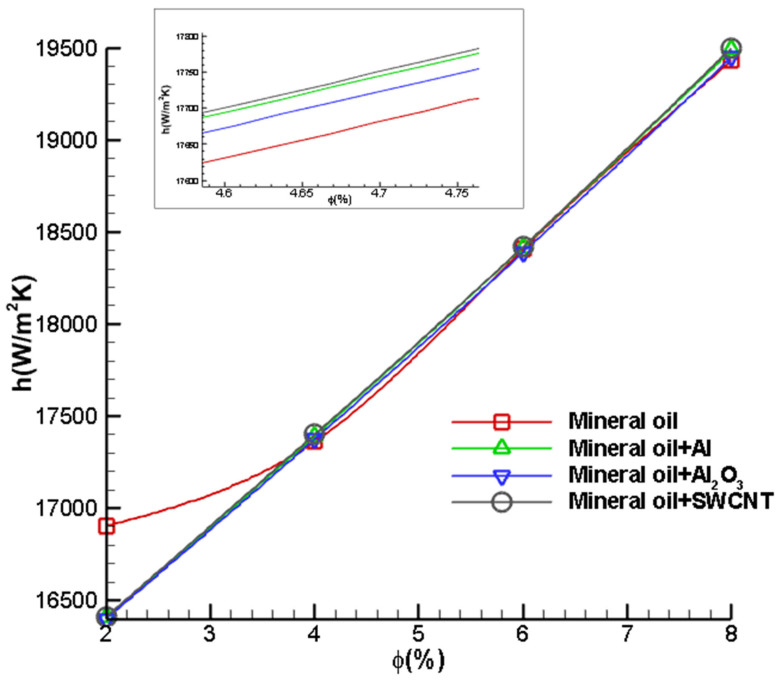
Variation in the convective heat transfer coefficient of the cutting tool with different volume fractions of dispersed nanoparticles.

**Figure 14 nanomaterials-12-03508-f014:**
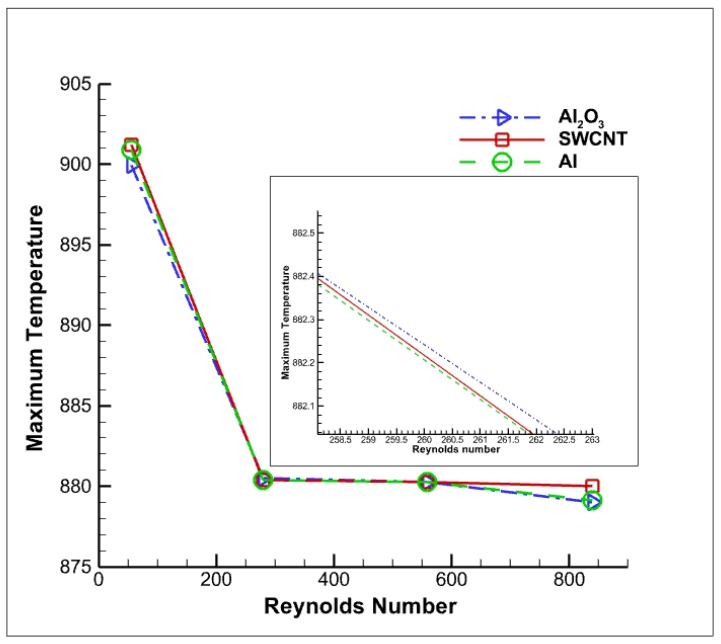
Variation in maximum cutting tool temperature with respect to Reynolds number.

**Figure 15 nanomaterials-12-03508-f015:**
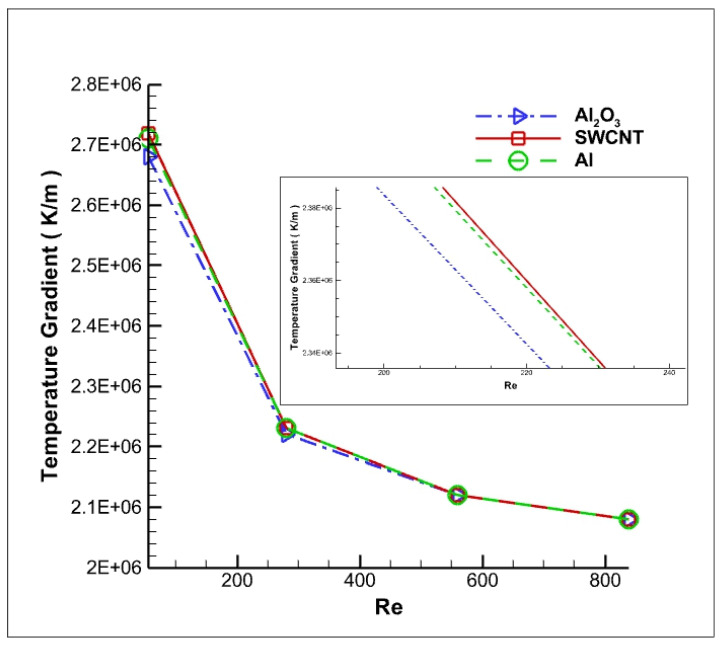
Variation in the temperature gradient of the cutting tool with respect to Reynolds number.

**Figure 16 nanomaterials-12-03508-f016:**
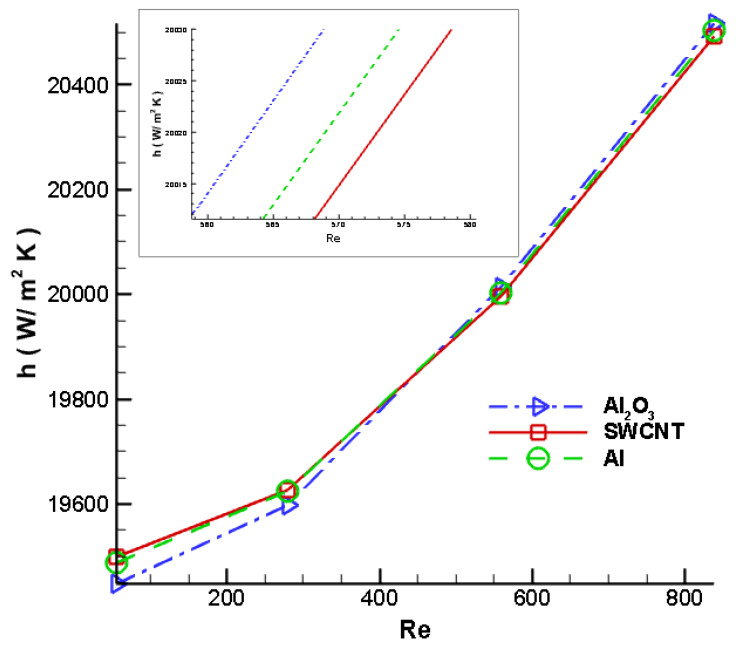
Variation in convective heat transfer coefficient of the cutting tool with respect to Reynolds number.

**Figure 17 nanomaterials-12-03508-f017:**
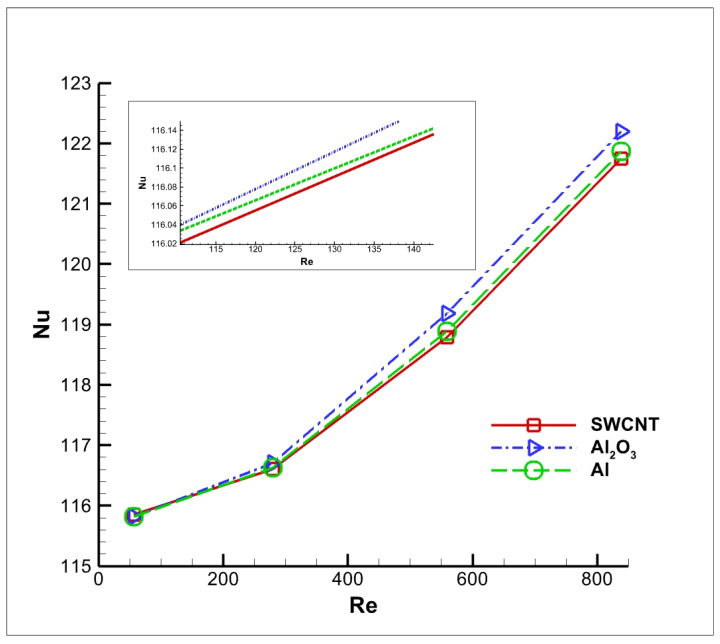
Variation in Nusselt number with respect to Reynolds number.

**Table 1 nanomaterials-12-03508-t001:** Thermophysical and mechanical properties of nanoparticles [43,44,45].

Nanoparticle	Thermal Conductivity (W/m-K)	Specific Heat (J/kg-K)	Density (kg/m^3^)
Al_2_O_3_	36.6	752	3970
SWCNT	6600	425	2600
Al	205	910	2700

**Table 2 nanomaterials-12-03508-t002:** Thermophysical and mechanical properties of mineral oil dispersed with Al_2_O_3_.

Cutting Fluid	Density (kg/m^3^)	Dynamic Viscosity (kg/m-s)	Thermal Conductivity (W/m-K)	Specific Heat (J/kg-K)
BASE FLUID - Mineral Oil	856.5	0.06380925	0.1335	1918
Mineral Oil + Al_2_O_3_ (2%)	918.77	0.071870783	0.141582886	1894.68
Mineral Oil + Al_2_O_3_ (4%)	981.04	0.080396228	0.149998749	1871.36
Mineral Oil + Al_2_O_3_ (6%)	1043.3	0.089385584	0.158768599	1848.04
Mineral Oil + Al_2_O_3_ (8%)	1105.6	0.098838852	0.167915248	1824.72

**Table 3 nanomaterials-12-03508-t003:** Thermophysical and mechanical properties of mineral oil dispersed with SWCNT.

Cutting Fluid	Density (kg/m^3^)	Dynamic Viscosity (kg/m-s)	Thermal Conductivity (W/m-K)	Specific Heat (J/kg-K)
BASE FLUID − Mineral Oil	856.5	0.06380925	0.1335	1918
Mineral Oil + SWCNT (2%)	891.37	0.069727418	0.141672963	1888.14
Mineral Oil + SWCNT (4%)	926.24	0.075905368	0.150186445	1858.28
Mineral Oil + SWCNT (6%)	961.11	0.082343099	0.15906218	1828.42
Mineral Oil + SWCNT (8%)	995.98	0.089040612	0.16832379	1798.56

**Table 4 nanomaterials-12-03508-t004:** Thermophysical and mechanical properties of mineral oil dispersed with Al.

Cutting Fluid	Density (kg/m^3^)	Dynamic Viscosity (kg/m-s)	Thermal Conductivity (W/m-K)	Specific Heat (J/kg-K)
BASE FLUID − Mineral Oil	856.5	0.06380925	0.1335	1918
Mineral Oil + Al (2%)	893.37	0.069883868	0.141657197	1897.84
Mineral Oil + Al (4%)	930.24	0.076233168	0.150153587	1877.68
Mineral Oil + Al (6%)	967.11	0.082857149	0.159010775	1857.52
Mineral Oil + Al (8%)	1004	0.089755812	0.168252241	1837.36

**Table 5 nanomaterials-12-03508-t005:** Thermal properties of cutting tool and workpiece.

Component	Density (kg/m^3^)	Thermal Conductivity (W/m-K)	Specific Heat (J/kg-K)
Workpiece— AISI 4130 Steel	7800	43	470
Cutting Tool— *(UNS T11302)*	8160	19	460

**Table 6 nanomaterials-12-03508-t006:** Maximum temperature attained by the cutting tool and workpiece.

Component	Kelvin (K)
Cutting Tool Temp	1003.15
Workpiece Temp	884.15

## Data Availability

Data available on request.

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
