# Peer review of "Computational Fluid Dynamics Simulation on Thermal Performance of Al/Al2O3/SWCNT Nanocoolants for Turning Operations"

_nanomaterials, 2022, doi:10.3390/nano12193508_

Round 1
Reviewer 1 Report
The manuscript entitled:
Comparison of thermal performance of Al/ Al₂O₃/SWCNT nanocoolants for turning operations
deals with a very interesting topic in the field of enhancing the thermal performance of cutting fluids via the dispersion of different nanofluids.
The manuscript is well designed and very good written with a comprehensive introduction part showing the state of the art in this field and justifying the necessity to carry out the presented work. There are, however, some minor revisions which must be addressed before suggesting the manuscript for publication in nanomaterials.
Ø Out of Tables 2-4, it is quite obvious that the properties of all three investigated nanofluids are similar, for example the enhancement of the thermal conductivity compared to the basic fluid vary between 6% at Phi=2% and 26% at Phi=8%. Accordingly it is expected that the difference between the performance of the three fluids should be marginally small. The good thing is that all results approve this fact which supports the CFD investigation carried out. The authors are asked, therefore, to put this fact as the main result and conclusion and to try to avoid the overinterpretation of the obtained results, especially if the three fluids shall be compared to each other.
Ø Please define for Figures 10 to Figure 13 the missed data concerning the velocity or the Re number of the nanofluid.
Ø Figure 10 is a good example of the over interpretation of the results. To the reader, all results are almost identical, so making use of lesser, significantly higher, higher an dmoderate shall be avoided. Instead, please be specific and quantify the relative performance between the three nanofluids.
Ø Figure 11 is another one showing a maximum deviation between all three fluids of 1 K at Phi=8%. Knowing that the temperature is 900 K, the difference is almost nothing. Increasing the volume fraction of the nanoparticles from 2 to 8 results of a temperature reduction of 6 to 7K (from 907 down to 900K). Again the relative temperature reduction is negligible (less than 0.8%), which is a result for itself, however, such a result must be expressed as is and in such a specific way.
Ø In all Figures 11 to 13, the simulation results of the basic nanofluids must be conducted and added to the figures. If the extrapolation of the presented results is allowed (therefore the necessity of adding the results of those simulations), the recommendation of applying nanofluids is, from the economic point of view, more than questionable. Again the results themselves are scientifically ok but the overmarketing of the obtained results should be avoided in a scientific paper.
Ø Figure 14 tells clearly that the increase of the Re number until ca. 260 has much bigger effect on the maximum temperature reduction (ca 22K). Further enhancement of Re number up to 850 has a much lower effect (ca. 1 K). A scientific interpretation for such a result must be given instead of concentrating on the differences between the three fluids, which is again negligible. The same is valid for Figure 15 for the temperature gradient.
Ø In general, a replication or an estimation of the lifetime increment of the cutting tools upon reducing the temperature by less than 0.8% should be given. If this is not possible, then please revise the whole conclusion section and remove all wordings in this direction. The conclusion section needs more specific values rather than marketing statements.
Very minor or editorial comments:
Page 1, first line of the Nomenclature: please correct the unit of h W m-2 K-1
Page 1, line 6 of the Nomenclature: Please unify the unit writing style and make use for Uc the unit (m s-1)
Page 4, line 9: the starting words of the sentence … the thermal conductivities … shall be added
Page 13, Capture of Figure 6: whether you delete the second “of” or add to it “the cutting tool and workpiece”
Author Response
Detailed response to Reviewer 1’s comments on the manuscript
I would like to thank the reviewer for taking the time and effort to provide
insightful guidance to improve the quality of our manuscript. I have considered all
the comments into consideration in the revised manuscript. The responses to the reviewer comments are appended below.
- Reviewer’s comments - Out of Tables 2-4, it is quite obvious that the properties of all three investigated nanofluids are similar, for example the enhancement of the thermal conductivity compared to the basic fluid vary between 6% at Phi=2% and 26% at Phi=8%. Accordingly, it is expected that the difference between the performance of the three fluids should be marginally small. The good thing is that all results approve this fact which supports the CFD investigation carried out. The authors are asked, therefore, to put this fact as the main result and conclusion and to try to avoid the overinterpretation of the obtained results, especially if the three fluids shall be compared to each other.
Author’s Response - The above statement i.e., the difference between the performance of the 3 nanofluids is relatively small, was incorporated as a main result in the conclusion section and the overinterpretation of the 3 results was avoided (Pg 27, Conclusion – Point a,b)
- Reviewer’s comments -Please define for Figures 10 to Figure 13 the missed data concerning the velocity or the Re number of the nanofluid.
Author’s Response – Figures 10-13 represent the variations in temperature, temperature gradient and heat transfer coefficient with respect to change in volume fraction of nanoparticles. All the simulations were carried out at a velocity of 1m/s (Re = 55). This point has been added in the starting paragraph of the results and discussions section for clarification and is constant to all the simulations where nanoparticle volume fraction was varied. (Pg 19 – Paragraph 1 – Line 3-5)
- Reviewer’s comments -Figure 10 is a good example of the over interpretation of the results. To the reader, all results are almost identical, so making use of lesser, significantly higher, higher, and moderate shall be avoided. Instead, please be specific and quantify the relative performance between the three nanofluids.
Author’s Response – The relative performance of nanofluids in terms of temperature distribution are quantified by comparing temperature at certain locations (Pg 18 – Paragraph 2)
- Reviewer’s comments -Figure 11 is another one showing a maximum deviation between all three fluids of 1 K at Phi=8%. Knowing that the temperature is 900 K, the difference is almost nothing. Increasing the volume fraction of the nanoparticles from 2 to 8 results of a temperature reduction of 6 to 7K (from 907 down to 900K). Again, the relative temperature reduction is negligible (less than 0.8%), which is a result for itself, however, such a result must be expressed as is and in such a specific way.
Author’s Response – The percentage reduction in maximum temperature was quantified and stated specifically in the results section. (Pg 27, Conclusion – Point b)
- Reviewer’s comments -In all Figures 11 to 13, the simulation results of the basic nanofluids must be conducted and added to the figures. If the extrapolation of the presented results is allowed (therefore the necessity of adding the results of those simulations), the recommendation of applying nanofluids is, from the economic point of view, more than questionable. Again, the results themselves are scientifically ok but the overmarketing of the obtained results should be avoided in a scientific paper.
Author’s Response – We have added the results for the case where we are using a basic cutting fluid. We have compared the respective values with cases where nanoparticles were added. (Pg 19-21 – Fig 11 to 13 have been updated to incorporate the results involving a base fluid simulation)
- Reviewer’s comments -Figure 14 tells clearly that the increase of the Re number until ca. 260 has much bigger effect on the maximum temperature reduction (ca 22K). Further enhancement of Re number up to 850 has a much lower effect (ca. 1 K). A scientific interpretation for such a result must be given instead of concentrating on the differences between the three fluids, which is again negligible. The same is valid for Figure 15 for the temperature gradient.
Author’s Response – The scientific interpretation and reasoning of the results has been stated for Figure 14-15 (Pg 22 – Paragraph 2 – Line 6 onwards till end)
- Reviewer’s comments -In general, a replication or an estimation of the lifetime increment of the cutting tools upon reducing the temperature by less than 0.8% should be given. If this is not possible, then please revise the whole conclusion section and remove all wordings in this direction. The conclusion section needs more specific values rather than marketing statements.
Author’s Response – We have edited the conclusion section and focused more on quantifying the increase/decrease in certain parameters chosen for analysis. The conclusions regarding the lifetime increment have been removed. (Pg 27, Conclusion – All points)
Very minor or editorial comments:
- Page 1, first line of the Nomenclature: please correct the unit of h W m-2 K-1
- Page 1, line 6 of the Nomenclature: Please unify the unit writing style and make use for Uc the unit (m s-1)
- Page 4, line 9: the starting words of the sentence … the thermal conductivities … shall be added
- Page 13, Capture of Figure 6: whether you delete the second “of” or add to it “the cutting tool and workpiece”
Author’s Response – The minor comments stated above were rectified and incorporated in the updated draft

Reviewer 2 Report
In this submission to Nanomaterials, the authors investigate the thermal performance of cutting fluids dispersed with nanoparticles for effective heat removal during turning operations. The authors find that nanocoolants with enhanced thermophysical properties are efficient in decreasing the maximum cutting temperature and improving the life of the cutting tool. The authors use finite element methods (further comments are given below) where the computational domain consists of a heated cutting tool and nanocoolants are sprayed from a nozzle located above the machining zone. The authors' results indicated a drastic drop in the cutting tool temperature with an increase in the volume fraction of dispersed nanoparticles and coolant velocity. The authors conclude that their numerical results are in good accordance with experimental results.
I find this manuscript to be of interest to researchers interested in modeling nanotube-based materials with finite element methods as well as readers of this journal. As such, I am somewhat supportive of publication with a few edits that should be incorporated in the next revision. In particular, while the authors use finite element methods for their calculations, there has been much work using atomistic methods (such as classical molecular dynamics) to understand heat and energy transfer in nanotubes, which should be noted:
Int. J. Heat Mass Transf. 2012, 55, 5007-5015
J. Chem. Theory Comput. 2011, 7, 1736–1749
In particular, these prior works showed that these classical molecular dynamics methods naturally give a higher resolution of what is happening in nanotube-material interfaces (compared to finite element methods). I am not necessarily asking the authors to carry out such types of calculations, but it should be mentioned that these other methods can give a higher resolution of heat transfer in these systems. With these minor revisions, I would be willing to re-review this manuscript for possible publication in Nanomaterials.
Author Response
Detailed response to Reviewer 2’s comments on the manuscript
I would like to thank the reviewer for taking the time and effort to provide
insightful guidance to improve the quality of our manuscript. I have considered all
the comments into consideration in the revised manuscript. The responses to the reviewer comments are appended below.
Reviewer’s comments - In this submission to Nanomaterials, the authors investigate the thermal performance of cutting fluids dispersed with nanoparticles for effective heat removal during turning operations. The authors find that nanocoolants with enhanced thermophysical properties are efficient in decreasing the maximum cutting temperature and improving the life of the cutting tool. The authors use finite element methods (further comments are given below) where the computational domain consists of a heated cutting tool and nanocoolants are sprayed from a nozzle located above the machining zone. The authors' results indicated a drastic drop in the cutting tool temperature with an increase in the volume fraction of dispersed nanoparticles and coolant velocity. The authors conclude that their numerical results are in good accordance with experimental results.
I find this manuscript to be of interest to researchers interested in modeling nanotube-based materials with finite element methods as well as readers of this journal. As such, I am somewhat supportive of publication with a few edits that should be incorporated in the next revision. In particular, while the authors use finite element methods for their calculations, there has been much work using atomistic methods (such as classical molecular dynamics) to understand heat and energy transfer in nanotubes, which should be noted:
Int. J. Heat Mass Transf. 2012, 55, 5007-5015
- Chem. Theory Comput. 2011, 7, 1736–1749
In particular, these prior works showed that these classical molecular dynamics methods naturally give a higher resolution of what is happening in nanotube-material interfaces (compared to finite element methods). I am not necessarily asking the authors to carry out such types of calculations, but it should be mentioned that these other methods can give a higher resolution of heat transfer in these systems. With these minor revisions, I would be willing to re-review this manuscript for possible publication in Nanomaterial.
Author’s Response – We have added the notion that classical molecular dynamics gives a more accurate resolution of nanoparticles dispersed in cutting fluids. These points were added in the introduction section citing the appropriate resources (Pg 5 – Paragraph 2 – Last 7 lines). References 35 and 36 were newly added.
